# Psychological Distress among University Staff before and during the COVID-19 Pandemic

**DOI:** 10.3390/ijerph20032208

**Published:** 2023-01-26

**Authors:** Takaki Tanifuji, Kentaro Mouri, Yasuji Yamamoto, Shinsuke Aoyama

**Affiliations:** 1Department of Psychiatry, Kobe University Graduate School of Medicine, Kobe 650-0017, Japan; 2Kobe University Inclusive Campus & Healthcare Center, Kobe 657-8501, Japan; 3Department of Internal Medicine, Division of Biosignal Pathophysiology, Kobe University Graduate School of Medicine, Kobe 650-0017, Japan

**Keywords:** mental health, psychological distress, COVID-19, SARS-CoV-2, occupational health, universities

## Abstract

(1) Background: The COVID-19 pandemic has distressed many populations worldwide, and since its beginning, many institutes have performed cross-sectional studies to assess mental health. We longitudinally examined psychological distress and depressive symptoms among university staff in Japan from 2019 to 2021, before and during the COVID-19 pandemic.; (2) Methods: Participants were teachers and hospital staff working at institutions related to Kobe University, who completed the Brief Job Stress Questionnaire (BJSQ) from 2019 to 2021. This study used the definition recommended by the guideline to identify high-stress. We analyzed the relationship between those who identified as having high-stress before versus during the COVID-19 pandemic using logistic regression analysis (adjusted for age, sex, and occupation).; (3) Results: Results showed that Stress Reaction scores increased slightly in 2020 and significantly in 2021. Time and other factors had a synergistic effect on mental health. The increase in Stress Reaction was significantly associated with females and nurses over the three years. Those with high-stress in 2019 had approximately twenty-fold odds ratios (OR) of having high-stress in 2020 and 2021.; (4) Conclusions: The long-term COVID-19 pandemic may disturb university staff’s mental health. Those who originally experienced high levels of stress were vulnerable to the negative effects of the COVID-19 pandemic.

## 1. Introduction

Since December 2019, repeated coronavirus disease-2019 (COVID-19) outbreaks have distressed populations worldwide due to lifestyle changes, social isolation, behavioral restrictions, and fear of infection. The prevalence of depression during the pandemic is seven times higher than the estimated global prevalence [1]. Mental health status is worse than before the COVID-19 pandemic in the Japanese population [2,3]. For the first time in 11 years, the number of suicide deaths in Japan has increased compared to before the pandemic [4].

Many studies on mental health during the COVID-19 pandemic have been conducted worldwide, resulting in many published systematic reviews [5,6,7,8,9,10,11,12,13,14,15]. Known risk factors for greater psychological distress include females’ sex, younger age, presence of mental illness, students, nurses, and a high risk of contracting COVID-19 [8,15]. The pandemic also greatly impacted the mental health problems of healthcare workers (HCWs) [16,17]. HCWs have a high prevalence of moderate or more depression [18,19,20], and a previous study demonstrated a high prevalence of depression (50.4%), anxiety (44.6%), insomnia (34%), and psychological distress (71.5%) due to the pandemic [18]. Females and nurses present a higher prevalence of affective symptoms than males and other HCWs [19,21,22]. Most of the studies included in these systematic reviews were cross-sectional and showed consistent results [5,7,8,10,11,14,15]. Few studies have compared mental conditions before and during the COVID-19 pandemic and conducted follow-up studies across the years.

Although mental health problems for HCWs have gathered attention, the COVID-19 pandemic has also negatively affected mental health in school-related populations [23,24,25]. Most studies have focused on students; few have examined teachers’ mental health [6,9,12,13]. According to a systematic review of teachers’ mental health, psychiatric symptoms vary between countries, highlighting the need for more research on the impact of the COVID-19 pandemic on teachers’ mental health [9]. Moreover, previous research has shown that the pandemic’s impact on mental health varies among teachers in elementary and junior high schools in Japan [26]; however, there is insufficient research investigating the mental health of Japanese university teachers. Wakui et al. suggested that administrative institutions should understand teachers’ psychological and physical stress using surveys because their mental health influences students’ education and development [26].

In a previous cross-sectional study, we investigated HCWs and students’ mental health during the first wave of the COVID-19 pandemic using the patient healthcare questionnaire-9 (PHQ-9), which demonstrated that nurses, females, and younger people were likely to have depression symptoms [19], which aligned with findings of previous research [8,15,21,22]. However, it was impossible to fully assess COVID-19’s impact on mental health due to the cross-sectional nature of that work. Therefore, we compare before and during the COVID-19 pandemic and investigate longitudinally to substantially uncover the pandemic’s long-term mental health effects. We hypothesized that nurses, females, and younger people have mental distress at workplaces before the COVID-19 pandemic and the long-term pandemic may exacerbate our mental health.

To address these gaps, we examined psychological distress and depressive symptoms in the workplace and changes over time among university staff, including teachers, HCWs, and office clerks, before and during the COVID-19 pandemic in Japan.

## 2. Materials and Methods

### 2.1. Study Design

This retrospective cohort study covers the period from 2019 to 2021, one year before and two years after the onset of the COVID-19 pandemic. We included participants working at institutions related to Kobe University who had undergone a stress check for all three years and agreed to participate in our study. We used the Brief Job Stress Questionnaire (BJSQ), which is performed annually for staff at Kobe University, to assess stress [27]. The BJSQ is the standard method for evaluating occupational stress and is recommended by the Japanese Ministry of Health, Labor and Welfare [27]. Information is gathered through online questionnaires and the distribution of questionnaires. All participants were given the opportunity to refuse to enroll in our study.

### 2.2. Participants

One person was omitted because of refusal. A total of 10,177 participants working at institutions related to Kobe University (including a hospital, combined junior and senior high school, special needs school, elementary school, and preschool) underwent a stress check that the workplace conducts annually, based on the Industrial Safety and Health Act, from 2019 to 2021 [27]. A total of 1609 participants (823 females and 786 males; mean age: 44.58 ± 11.02 years in 2021) underwent the stress check all three years. The BJSQ is performed annually in November. We classified workers into six categories: office clerks, teachers, nurses, allied health professionals, doctors, and others, in accordance with the departments and workplace. We defined medical teachers in clinical fields as doctors. Of all the teachers (488 [100%]), most were university teachers (432 [89%]), and the rest were combined junior and senior high school (29 [6%]), special needs school (17 [3%]), elementary school (7 [1%]), and preschool (3 [1%]). Allied health professionals included occupational therapists, physical therapists, speech therapists, orthoptists, pharmacists, clinical psychologists, radiological technologists, medical technologists, clinical engineering technologists, registered dietitians, and dental hygienists. Other included mostly part-time workers.

### 2.3. Measures

The BJSQ was developed by Shimomitsu and authorized by the Ministry of Health, Labor and Welfare for implementing stress checks in workplaces across Japan; previous studies have demonstrated acceptable reliability and validity [27,28,29,30,31]. The BJSQ has been used in various studies [32,33,34,35,36]. It contains 57 items scored on a four-point Likert-type scale, and items are largely categorized into three components: Job Stressor (items 1−17, 17−68 points), Stress Reaction (items 18−46, 29−116 points), and Social Support (items 47−55, 9−36 points); items 56 and 57 are excluded because they are not used on the stress check [29,37] (Supplementary, BJSQ English version). Higher total scores for each category indicate higher stress and poorer social support. Job Stressor comprises nine subscales: quantity and quality of psychological job demands, physical burden, interpersonal relationships, work environment, job control, skill utilization, suitability of work, and meaningfulness of work. Stress Reaction comprises six subscales: lack of liveliness, irritability, fatigue, anxiety, depression, and somatic symptoms. Three of these categories are considered when diagnosing major depressive disorder according to the Diagnostic and Statistical Manual of Mental Disorders, 5th Edition (DSM-5). We posited that the condition in which Stress Reaction scores are high and psychological distress is exacerbated may be indicative of depression. Social Support comprises three subscales: support from superiors, co-workers, and family/friends. The depression subscale in Stress Reaction comprises the scores of items 30 to 35 (6−24 points) [29,37] (Supplementary, BJSQ English version).

The Ministry of Health, Labor and Welfare guideline recommends the following criteria to define high-stress: a Stress Reaction of 77 points or higher; or the sum of Job Stressor and Social Support of 76 points or higher, in addition to a Stress Reaction of 63 points or higher [29]; this study used the same definition. High-stress is considered to have a sensitivity of 60.5% and specificity of 89.9% when screening for psychological distress, defined as a Kessler 6 scale of more than 13 points [30]. Those diagnosed with high-stress by the BJSQ were at risk of taking long-term sickness absence for 6.59 hazard ratios (HR) in males and 2.77 HR in females [31].

### 2.4. Statistical Analysis

Statistical analyses were performed using R version 4.0.3 (R Development Core Team, Vienna, Austria) with EZR version 1.54 (Jichi Medical University, Saitama, Japan) [38] and SPSS version 26.0 (IBM Corp, USA). The continuous variables for identical participants over three years were analyzed using repeated measures of one- or two-way ANOVA (factors: sex, occupation, or age in 2021); two-way ANOVA indicated interaction and main effect of *p*-values, respectively, and post-hoc analysis was performed using the Bonferroni method. Categorical variables were analyzed using Fisher’s exact test. We performed multiple linear regression analysis to address confounding factors (adjusted for age, sex, occupation, and [years]). We analyzed the relationship between those identified as having high-stress in 2019 and 2020 or 2021 using logistic regression analysis (adjusted for age in 2021, sex, and occupation). We defined two-tailed *p*-value < 0.05 as statistical significance. Dummy variables were used, as necessary (occupation, office clerk = 0 and all other occupations = 1; sex, male = 0 and female = 1; [year, 2019 = 0 and all other years = 1]).

## 3. Results

Table 1 summarizes the demographic data and the number of high-stress populations.

### 3.1. Stress Reaction, Job Stressor, and Social Support

The Stress Reaction category of the BJSQ and its Depression subscale showed a similarly increasing tendency each year, and figures based on each factor (including sex, occupation, and age in 2021) mostly indicated the same tendency (Figure 1). The scores of Stress Reaction in 2021 have increased significantly compared with 2019 and 2020; the results are similarly based on each factor (Appendix A). Other categories of the BJSQ are described in Appendix A. The interaction of Stress Reaction showed a significant difference for all factors (Time/sex, *p* = 0.011; Time/age, *p* < 0.001; Time/occupation, *p* < 0.001) (Appendix A). The main effects of each Time and factor were significantly different in terms of Stress Reaction (Time, *p* < 0.001; Time and sex, Time, *p* < 0.001, sex, *p* < 0.001; Time and age, Time, *p* < 0.001, age, *p* < 0.001; Time and occupation, Time, *p* < 0.001, occupation, *p* < 0.001) (Appendix A). All interactions of Job Stressor differed significantly; all main effects of each Time and factor in Job Stressor were also significantly different (Time, *p* < 0.001; Time and sex, Time/sex, *p* < 0.001, Time, *p* < 0.001, sex, *p* = 0.002; Time and age, Time/age, *p* < 0.001, Time, *p* < 0.001, age, *p* < 0.001; Time and occupation, Time/occupation, *p* < 0.001, Time, *p* = 0.011, occupation, *p* < 0.001) (Appendix A). The interaction of Social Support showed a significant difference in Time/occupation (*p* = 0.013) (Appendix A). The main effects of each Time and age were significantly different in Social Support (Time, *p* < 0.001; Time and sex, Time, *p* < 0.001; Time and age, Time, *p* < 0.001, age, *p* < 0.001; Time and occupation, Time, *p* < 0.001) (Appendix A).

### 3.2. Factors Associated with Stress at the Workplace

Following adjustment for confounding factors (age, sex, and occupation), the increase in scores of Stress Reaction was significantly associated with females and nurses over three years, before and during the COVID-19 pandemic(females, *p* < 0.001 in 2019, *p* < 0.001 in 2020, *p* < 0.001 in 2021; nurses, *p* = 0.012 in 2019, *p* < 0.001 in 2020, *p* < 0.001 in 2021) (Table 2). Teachers, nurses, allied health professionals, and doctors were related to elevated scores of Job Stressor during the 19 pandemic (teachers, *p* < 0.001 in 2020, *p* = 0.022 in 2021; nurses, *p* < 0.001 in 2020, *p* < 0.001 in 2021; allied health professional, *p* < 0.001 in 2020, *p* < 0.001 in 2021; doctors, *p* < 0.001 in 2020, *p* = 0.001 in 2021) (Table 2). Higher age led to an increase in scores of Social Support, despite the pandemic (higher age, *p* < 0.001 in 2019, *p* < 0.001 in 2020, *p* < 0.001 in 2021) (Table 2).

Appendix A shows the risk factors associated with the three components of the BJSQ. After adjusting for confounding factors (age, sex, occupation, and years), increased scores of Stress Reaction were associated significantly with females, nurses, and allied health professionals both before and during the COVID-19 pandemic (*p* < 0.001, *p* < 0.001, *p* = 0.046, respectively) (Appendix A). Teachers, nurses, allied health professionals, and doctors were associated with increased scores of Job Stressor; the relation was stronger for HCWs than teachers, based on the standard partial regression coefficient (teachers [the standard partial regression coefficient = 0.0698/*p*-value < 0.001); nurses (0.3282/*p* < 0.001); allied health professionals (0.1465/*p* < 0.001); doctors (0.0856/*p* < 0.001)) (Appendix A). Social Support’s standard partial regression coefficient for teachers was less than other occupations (teachers (the standard partial regression coefficient = − 0.0434/*p*-value = 0.008); nurses (0.0115/*p* = 0.480); allied health professionals (0.0370/*p* = 0.011); doctors (−0.0231/*p* = 0.115)) (Appendix A). Older age was found to be associated with decreases in scores of Stress Reaction and Job Stressor as well as an increase in scores of Social Support (Stress Reaction (the standard partial regression coefficient = − 0.0020/*p*-value = 0.002); Job Stressor (−0.0014/*p* = 0.030); Social Support (0.0072/*p* < 0.001)) (Appendix A).

### 3.3. High Stress at the Workplace before and during the Pandemic

Those who identified as having high-stress in 2019 had approximately a twenty-fold higher OR for being diagnosed with high-stress in 2020 and 2021 than those who were not, after controlling for age, sex, and occupation (adjusted OR = 22.00, *p* < 0.001; adjusted OR = 17.60, *p* < 0.001, respectively) (Table 3). The number of staff diagnosed with high-stress increased in 2021 significantly compared with 2019 (2019 vs. 2020, 196 (12.2%) vs. 191 (11.9%), *p* = 0.828; 2019 vs. 2021, 196 (12.2%) vs. 259 (16.1%), *p* = 0.002) (Table 3).

## 4. Discussion

To the best of our knowledge, this is the first study examining the long-term effects of mental health over three years before and during the COVID-19 pandemic among the same group of university staff in Japan. All staff had some increase in distress during the COVID-19 pandemic compared with the pre-pandemic period; the second year of the pandemic affected mental health more significantly than before the pandemic and the first year afterward. Time and factors including sex, occupation, and age synergistically affected mental health; in particular, increased Stress Reaction was related to female sex and nurses over the three years. Older people tended not to benefit from surroundings support compared to younger people. Those with high psychological distress before the COVID-19 pandemic were at approximately a twenty-fold risk of experiencing high psychological distress during the pandemic compared with those who did not have high psychological distress beforehand.

There was a slight increase in mental health problems soon after the outbreak of the COVID-19 pandemic, which was reduced approximately to pre-pandemic levels by the middle of 2020 [39]. Mental health problems peaked in April and May 2020, and depressive and anxiety symptoms decreased over time [40]; however, mental health problems remain high during the COVID-19 pandemic [40]. In addition, psychological distress effects varied significantly by population from the two months after the pandemic [41]. There were relatively few studies in 2021, limiting a definitive understanding of the longitudinal effects on mental health in Japan; our results showed that psychological distress increased slightly in 2020 compared to 2019 and significantly in 2021 compared to 2019–2020. The long duration of the COVID-19 pandemic may have had negative effects on the mental health of university staff.

We found that nurses and females felt more distress, including depressive symptoms, than other occupations and males. This was also the case before the COVID-19 pandemic, although it worsened afterward. A global study showed that the prevalence of major depression was higher in females than males before the COVID-19 pandemic [42]. Moreover, several studies showed that the prevalence of depressive symptoms during the COVID-19 pandemic was higher in females and nurses than among males and doctors [19,21,22]. Studies prior to the pandemic showed that the prevalence of depressive symptoms in nurses was relatively high; 38% in China and 18% in the USA [43,44], and the prevalence of depression was 10% in Canada [45]. Those employed in roles that involve interaction with the general public are at an elevated risk of psychological distress during the COVID-19 pandemic [46]. In Japan, females are frequently engaged in the hospitality sector, including caregivers, nurses, reception workers, and cashiers; these kinds of gender roles may impact mental health. Our findings also indicated that individuals with high psychological distress before the pandemic were approximately twenty-fold more vulnerable to distress during the pandemic. This suggests that the pandemic may have deteriorated the mental condition of those who already had distress. We emphasize the importance of regularly performing stress checks at work and handling the matter as necessary.

Older people did not feel the benefit of surroundings support from superiors, co-workers, family, and friends compared to young people. Emotional loneliness has been linked to depression in older adults [47]. However, this present study showed that young people were more likely to feel distressed, including experiencing depressive symptoms than older people, which is supported by previous studies [15,19]. It might be that other factors worsened the university staff’s psychological condition, or that young people who became more stressed may have asked for social support. Our results showed that teachers were more related to a decrease in Social Support’s points than other occupations, including nurses, based on the standard partial regression coefficient; teachers may make more efficient use of surroundings supports compared to nurses. Our findings could not conclude that higher stress was related directly to an increase in the demand for support from people around. Further studies are required to elucidate the relationship between age, Social Support, and workplace distress during the pandemic.

This study has several limitations. First, we did not consider social situations, including the waves of the pandemic, lockdown, and variants of COVID-19. The stress check was performed in 2020, the early stage of the third wave, and in 2021 after the fifth wave in Japan. Second, this study used self-report questionnaires and relied on subjectivity. The BJSQ was developed and validated for stress checks in the workplace and is not a specific questionnaire for major depressive disorder. Third, we could not examine mental health status before 2019. Thus, we used only one year of data from 2019, compared to the two years during the COVID-19 pandemic. Fourth, our results did not indicate the mental status of the general population. Our participants included teachers and HCWs, which may have led to biased populations. Finally, we did not collect detailed characteristics, including their household status, income, and medical history.

## 5. Conclusions

Our results showed that the psychological distress of university staff increased slightly in 2020 and significantly in 2021. The long duration of the COVID-19 pandemic may disturb mental health. Time and various factors such as sex, occupation, and age had synergistic effects on mental health. Females and nurses were associated with psychological distress before the pandemic. Those who originally felt distressed were vulnerable to the negative effects of the pandemic on mental health. While carefully monitoring the impact of a prolonged pandemic on mental health, each workplace needs to consider specific interventions for those with high stress during the pandemic.

## Figures and Tables

**Figure 1 ijerph-20-02208-f001:**
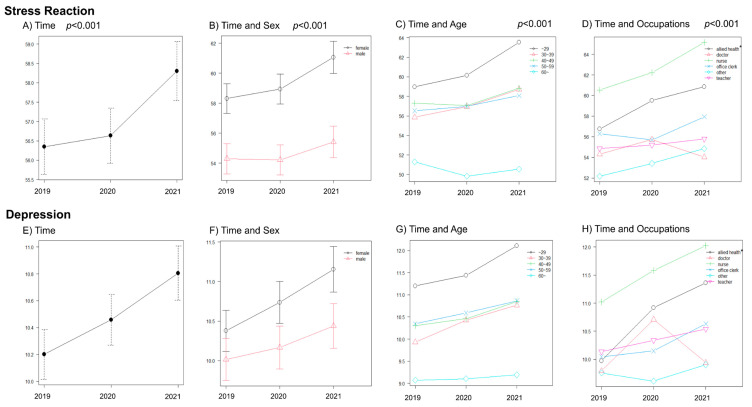
Stress Reaction and Depression among university staff before and during the COVID-19 pandemic. Stress Reaction is one component of the BJSQ and is subdivided into six categories: lack of liveliness, irritability, fatigue, anxiety, depression, and somatic symptoms. Depression scores are a category extracted from six subdivided ones. Stress Reaction and Depression demonstrate (**A**–**D**) and (**E**–**H**), respectively. (**A**)/(**E**) Time, (**B**)/(**F**) Time and Sex, (**C**)/(**G**) Time and Age, and (**D**)/(**H**) Time and Occupation. Line graphs demonstrate changes in Stress Reaction or Depression scores over time, and *p*-values of Time’s main effects were obtained using repeated measures of two-way ANOVA. Error bars indicate 95% CI. * allied health = allied health professional ANOVA, analysis of variance; BJSQ, brief job stress questionnaire; CI, confidence interval; COVID-19, coronavirus disease-2019.

**Table 1 ijerph-20-02208-t001:** Demographic Characteristics of university staff.

Group	Individuals by Occupation, Number (%)						
	Office Clerk	Teacher	Nurse	Allied Health Professional ^a^	Doctor	Other	Total
**occupation**	719 (44.7)	488 (30.3)	241 (15.0)	72 (4.5)	48 (3.0)	41 (2.5)	1609 (100)
**sex**							
female	421 (58.6)	117 (24.0)	223 (92.5)	41 (56.9)	6 (12.5)	15 (36.6)	823 (51.1)
male	298 (41.4)	371 (76.0)	18 (7.5)	31 (43.1)	42 (87.5)	26 (63.4)	786 (48.9)
**total**	719 (100)	488 (100)	241 (100)	72 (100)	48 (100)	41 (100)	1609 (100)
**age (SD) ^b^**	44.9 (10.08)	49.9 (9.62)	35.0 (9.86)	39.53 (8.54)	46.96 (7.52)	38.17 (12.49)	
**age/generation**							
~29	52 (7.2)	7 (1.4)	98 (40.7)	7 (9.7)	0 (0.0)	9 (22.0)	173 (10.75)
30–39	180 (25.0)	71 (14.5)	76 (31.5)	31 (43.1)	7 (14.6)	19 (46.3)	384 (23.87)
40–49	239 (33.2)	143 (29.3)	37 (15.4)	26 (36.1)	27 (56.2)	4 (9.8)	476 (29.58)
50–59	189 (26.3)	179 (36.7)	29 (12.0)	7 (9.7)	9 (18.8)	6 (14.6)	419 (26.04)
60~	59 (8.2)	88 (18.0)	1 (0.4)	1 (1.4)	5 (10.4)	3 (7.3)	157 (9.76)
**total**	719 (100)	488 (100)	241 (100)	72 (100)	48 (100)	41 (100)	1609 (100)
**High-stress population**							
2019	88 (12.2)	51(10.5)	41 (17.0)	9 (12.5)	3 (6.2)	4 (9.8)	196 (12.2)
2020	74 (10.3)	57 (11.7)	41 (17.0)	12 (16.7)	2 (4.2)	5 (12.2)	191 (11.9)
2021	101 (14.0)	62 (12.7)	70 (29.0)	18 (25.0)	4 (8.3)	4 (9.8)	259 (16.1)

^a^ occupational therapists, physical therapists, speech therapists, orthoptists, pharmacists, clinical psychologists, radiological technologists, medical technologists, clinical engineering technologists, registered dietitians, and dental hygienists. ^b^ the age indicated mean and SD in 2021. BJSQ, brief job stress questionnaire; SD, standard deviation.

**Table 2 ijerph-20-02208-t002:** Multiple regression analyses of university staff before and during the COVID-19 pandemic each year.

	Year	Sex	Age	Teacher	Nurse	Allied Health Professional ^a^	Doctor
Stress Reaction							
	2019	**0.09981228 *****	−0.02578665	−0.00750750	**0.07105174 ***	0.00461643	−0.00651759
		*p* < 0.001	*p* = 0.354	*p* = 0.792	*p* = 0.012	*p* = 0.856	*p* = 0.798
	2020	**0.12174035 *****	−0.04374771	0.03216826	**0.11609670 *****	**0.05054031 ***	0.02087680
		*p* < 0.001	*p* = 0.113	*p* = 0.255	*p* < 0.001	*p* = 0.0452	*p* = 0.408
	2021	**0.11108758 *****	**−0.07482003 ****	−0.01204604	**0.11447505 *****	0.03206125	−0.02293297
		*p* < 0.001	*p* = 0.006	*p* = 0.668	*p* < 0.001	*p* = 0.200	*p* = 0.360
Job Stressor							
	2019	**−0.05838221 ***	0.02953820	0.05138921	**0.31062047 *****	**0.14295018 *****	**0.08090630 ****
		*p* = 0.032	*p* = 0.272	*p* = 0.063	*p* < 0.001	*p* < 0.001	*p* = 0.001
	2020	0.00791290	−0.04330850	**0.09650523 *****	**0.34007493 *****	**0.15301932 *****	**0.09955577 *****
		*p* = 0.766	*p* = 0.097	*p* < 0.001	*p* < 0.001	*p* < 0.001	*p* < 0.001
	2021	−0.00872748	**−0.083301800 ****	**0.06143698 ***	**0.33392813 *****	**0.14373674 *****	**0.07658097 ****
		*p* = 0.742	*p* = 0.002	*p* = 0.022	*p* < 0.001	*p* < 0.001	*p* = 0.001
Social Support							
	2019	−0.01858601	**0.18246364 *****	**−0.05880193 ***	−0.02413241	0.03877993	−0.03082414
		*p* = 0.507	*p* < 0.001	*p* = 0.038	*p* = 0.392	*p* = 0.126	*p* = 0.224
	2020	−0.00258984	**0.16842258 *****	−0.04545781	0.01772081	0.04854749	−0.02290526
		*p* = 0.927	*p* < 0.001	*p* = 0.111	*p* = 0.532	*p* = 0.056	*p* = 0.368
	2021	−0.01023959	**0.15732397 *****	−0.02635865	0.04085243	0.02404848	−0.01579599
		*p* = 0.717	*p* < 0.001	*p* = 0.357	*p* = 0.151	*p* = 0.346	*p* = 0.536

^a^ occupational therapists, physical therapists, speech therapists, orthoptists, pharmacists, clinical psychologists, radiological technologists, medical technologists, clinical engineering technologists, registered dietitians, and dental hygienists. We demonstrate the standard partial regression coefficient. Dummy variables: occupation, office clerk = 0, all other occupations = 1; sex, male = 0 and female = 1. * *p*-value < 0.05, ** *p*-value < 0.01, *** *p-*value < 0.001. The standard partial regression coefficient, denoted in bold, is significant at <0.05.

**Table 3 ijerph-20-02208-t003:** Analysis of high-stress population proportion before and during the COVID-19 pandemic.

Year	High-Stress Population Number (%)	Adjusted OR (95% CI) ^a^	*p*-Value
2019/2020 ^b^	196 (12.2)/191 (11.9)		0.828
2020		**22.00 (15.20−31.8) *****	<0.001
2019/2021 ^b^	196 (12.2)/259 (16.1)		0.002
2021		**17.60 (12.3−25.0) *****	<0.001

^a^ We demonstrate analysis between those who were identified as having high-stress in 2019 and 2020 or 2021 using logistic regression analysis (adjusted for age in 2021, sex, and occupation). ^b^ We assessed the proportion between 2019, and 2020 or 2021 using Fisher’s exact test. *** *p*-value < 0.001. The OR denoted in bold is significant at <0.05. CI, confidence interval; OR, odds ratio.

## Data Availability

The raw data related to this article are subject to the following restrictions: data used for secondary purposes must be approved by the Ethical Committee of the Kobe University Graduate School of Medicine. Requests to access these datasets should be directed to the corresponding author.

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
