# Peer review of "Psychological Distress among University Staff before and during the COVID-19 Pandemic"

_ijerph, 2023, doi:10.3390/ijerph20032208_

Round 1

Reviewer 1 Report

Review of “Psychological distress among university staff before and during the COVID-19 pandemic” 

In general, this was a sound manuscript and I enjoyed reading it. The only semi-major addition that I would like to see is a correlation matrix with age, sex, and the BJSQ scales for the first test period in the lower diagonal and in the third (final) test year in the upper diagonal. The purpose behind this request is to establish that the relationships between the variables themselves did not change over time. 

More minor comments: 

Line 40 – finish the sentence explaining that “due to the pandemic”. If not, then the sentence suggests that health care workers are depressed. 

Line 63 – remove the word “must” (not necessary here). 

Line 303 - “wav” should be “wave”. 

Author Response

We greatly appreciate the review of our paper entitled “Psychological distress among university staff before and during the COVID-19 pandemic”; Manuscript ID: ijerph-2179558. The comments of the reviewers were helpful in improving and polishing our paper. We have revised our paper extensively according to the comments and marked up any revisions in the manuscript using the “Track Changes” function for better readability. Point by point systematic replies to the reviewers are below.

Reviewers' comments:

Reviewer #1: In general, this was a sound manuscript and I enjoyed reading it. The only semi-major addition that I would like to see is a correlation matrix with age, sex, and the BJSQ scales for the first test period in the lower diagonal and in the third (final) test year in the upper diagonal. The purpose behind this request is to establish that the relationships between the variables themselves did not change over time.

Response: We appreciate of the reviewers’ positive remarks and constructive feedback. Our study includes participants who underwent the stress check in all three years, and the subject of the present study was the same individual. Regarding sex, we have only 2021 data. We may not have accurately inferred your intent, although. We have analyzed a correlation matrix with age, sex, and the BJSQ scales you pointed out using spearman’s rank correlation coefficient as follows:

BJSQ

2021

-0.125

< 0.001

0.147

< 0.001

0.723

< 0.001

-0.125

< 0.001

0.147

< 0.001

0.801

0

-0.125

< 0.001

0.147

< 0.001

1

0

sex

2021

-0.267

< 0.001

1

0

0.0922

< 0.001

-0.267

< 0.001

1

0

0.131

< 0.001

-0.267

< 0.001

1

0

0.147

< 0.001

age

2021

1

0

-0.267

< 0.001

-0.037

0.144

1

0

-0.267

< 0.001

-0.0912

< 0.001

1

0

-0.267

< 0.001

-0.125

< 0.001

BJSQ

2020

-0.091

< 0.001

0.131

< 0.001

0.766

< 0.001

-0.0912

< 0.001

0.131

< 0.001

1

0

-0.0912

< 0.001

0.131

< 0.001

0.801

0

sex

2020

-0.267

< 0.001

1

0

0.0922

< 0.001

-0.267

< 0.001

1

0

0.131

< 0.001

-0.267

< 0.001

1

0

0.147

< 0.001

age

2020

1

0

-0.267

< 0.001

-0.037

0.144

1

0

-0.267

< 0.001

-0.0912

< 0.001

1

0

-0.267

< 0.001

-0.125

< 0.001

BJSQ

2019

-0.037

0.144

0.0922

< 0.001

1

0

-0.037

0.144

0.0922

< 0.001

0.766

< 0.001

-0.037

0.144

0.0922

< 0.001

0.723

< 0.001

sex

2019

-0.267

< 0.001

1

0

0.0922

< 0.001

-0.267

< 0.001

1

0

0.131

< 0.001

-0.267

< 0.001

1

0

0.147

< 0.001

age

2019

1

0

-0.267

< 0.001

-0.037

0.144

1

0

-0.267

< 0.001

-0.091

< 0.001

1

0

-0.267

< 0.001

-0.125

< 0.001

age

2019

sex

2019

BJSQ

2019

age

2020

sex

2020

BJSQ

2020

age

2021

sex

2021

BJSQ

2021

The correlation matrix shows the inter-correlations (Spearman’s rho, p-value).

Dummy variables: sex, male = 0 and female= 1.

Major Comments

Line 40 – finish the sentence explaining that “due to the pandemic”. If not, then the sentence suggests that health care workers are depressed.

Line 63 – remove the word “must” (not necessary here).

Line 303 - “wav” should be “wave”.

Response and modification: We thank the reviewer for the valuable comment and for pointing out a mistake. We have made revisions throughout the manuscript on the basis of your advice.

- HCWs have a high prevalence of moderate depression [8], with approximately one in four presenting with depression due to the pandemic [9].

(Page1, line 42)

- Therefore, we compare before and during the COVID-19 pandemic and investigate longitudinally to substantially uncover the pandemic’s long-term mental health effects.

(Page2, line 65)

- The stress check was performed in 2020, the early stage of the third wave, and in 2021 after the fifth wave in Japan.

(Page12, line 333)

Reviewer 2 Report

The manuscript analyses a topic of vital importance because of the impact of the pandemic on people's mental health. The text presents exciting and convincing results. However, I have some doubts that need to be clarified before publishing the article:

1. Could you tell me why study professionals from that university are behind? Are these the only data available?

2. The authors quote several systematic reviews; I recommend directly quoting the results of the original articles of these reviews.

3. Given that this is a statistical analysis, is it necessary to start with research hypotheses?

4. What do you mean by psychological distress? I suggest conceptualising the main variable of the study

5. If R and SPSS serve the same purpose, why did you use both, and for what data were both used?

6. Do the rows or columns add up to 100% in the tables? I suggest adding a row or column for totals to make it easier to understand the %.

7. In Table 1, at the end (high-stress population), should each year add up to 100%? In 2021 they almost added up, but in 2019 and 2020, they did not.

8. The study's results should be "discussed" in more depth in the Discussion. Don't just quote other similar studies. For example, why do you think women or nurses are more affected? Why is there a difference in the age-related effect? Why is it that women are more affected, why is it that nurses are more affected, and why is there an age difference? Could you explain these differences with your ideas and quote research that has described it?

9. Check word on line 303

10. The conclusions seem straightforward; they should answer the question: has examining psychological distress only allowed you to discover that it has increased with the pandemic? What other data did you see?

In closing, Figure 1 allows us to see the differences per year in each variable more clearly.

Author Response

We greatly appreciate the review of our paper entitled “Psychological distress among university staff before and during the COVID-19 pandemic”; Manuscript ID: ijerph-2179558. The comments of the reviewers were helpful in improving and polishing our paper. We have revised our paper extensively according to the comments and marked up any revisions in the manuscript using the “Track Changes” function for better readability. Point by point systematic replies to the reviewers are below.

Reviewers' comments:

Reviewer #2: The manuscript analyses a topic of vital importance because of the impact of the pandemic on people's mental health. The text presents exciting and convincing results. However, I have some doubts that need to be clarified before publishing the article:

Response: We appreciate the reviewer’s constructive comments and raising some important points that should be addressed. Point by point replies for the reviewer are below.

  1. Could you tell me why study professionals from that university are behind? Are these the only data available?

Response: We procured the data during the preceding summer and utilized all available data obtained from the subcontractors outsourced by the General Affairs Department. As of yet, we do not possess data for the year 2022. One individual was omitted because of refusal. A total of 10,177 participants working at institutions related to Kobe University underwent a stress check that the workplace conducts annually from 2019 to 2021. A total of 1,609 participants underwent the stress check in all three years. It is plausible that there may be a certain number of individuals who did not undergo stress examinations. However, we currently lack data on response rates.

  1. The authors quote several systematic reviews; I recommend directly quoting the results of the original articles of these reviews.

Response and Modification: We appreciate the reviewers’ valuable comments. Systematic reviews have been cited as deemed necessary, and we have added the appropriate references on the basis of your suggestion.

  1. Given that this is a statistical analysis, is it necessary to start with research hypotheses?

Response and Modification: We thank the reviewer for the valuable comment and have added our hypotheses in the introduction section as follows:

- We hypothesized that nurses, females, and younger people have mental distress at workplaces before COVID-19 pandemic and the long-term pandemic may exacerbate our mental health.

(Page 2, lines 67-69)

  1. What do you mean by psychological distress? I suggest conceptualising the main variable of the study

Response and Modification: It is posited that psychological distress is akin to depression. The Stress Reaction in the BJSQ has been divided into six categories as follows: lack of liveliness, irritability, fatigue, anxiety, depression, and somatic symptoms. Three of these categories are considered when diagnosing major depressive disorder according to the Diagnostic and Statistical Manual of Mental Disorders, 5th Edition (DSM-5). We have added the following text in the Materials and Methods (2.3. Measures).

- Three of these categories are considered when diagnosing major depressive disorder according to the Diagnostic and Statistical Manual of Mental Disorders, 5th Edition (DSM-5). We posited that the condition in which Stress Reaction scores are high and psychological distress is exacerbated may be indicative of depression.

(Page 3, lines 116-120)

  1. If R and SPSS serve the same purpose, why did you use both, and for what data were both used?

Response: A repeated measures two-way analysis of variance was conducted to evaluate the scores of Social Support for each year's comparisons using EZR (R). However, EZR was unable to demonstrate the analysis results of multiple comparisons for Time and Time/sex in Social Support (Table S3). We verified it using SPSS, while all other analyses were performed using EZR.

  1. Do the rows or columns add up to 100% in the tables? I suggest adding a row or column for totals to make it easier to understand the %.

Response and Modification: We are grateful to the reviewer for their insightful suggestion. The rows or columns in the tables sum up to 100%. We agree with the reviewers’ comment that we add a row or column for totals to make it easier to understand the % because Table 1 may perplex readers. We amended Table 1, which in turn facilitated the understanding of Table 3.

  1. In Table 1, at the end (high-stress population), should each year add up to 100%? In 2021 they almost added up, but in 2019 and 2020, they did not.

Response: This question may be linked to your question of No 6 and I’m sorry Table 1 confused you. The row labeled “High-stress population” indicates the proportion of the total; for example, in the column of an office clerk, 88/719 in 2019, 74/719 in 2020, and 101/719 in 2021; in the row of 2019, 88/719 in office clerk, 51/488 in teacher, 41/241 in nurse, 9/72 in Allied health Professional, 3/48 in doctor, 4/41 in other, 196/1609 in total. We revised Table 1 not to mislead readers through your advice of No 6.

  1. The study's results should be "discussed" in more depth in the Discussion. Don't just quote other similar studies. For example, why do you think women or nurses are more affected? Why is there a difference in the age-related effect? Why is it that women are more affected, why is it that nurses are more affected, and why is there an age difference? Could you explain these differences with your ideas and quote research that has described it?

Response and Modification: We thank the reviewer for raising this important point. According to the suggestion, we have added the manuscript in the discussion section with an appropriate reference as follows:

- Those employed in roles that involve interaction with the general public are at an elevated risk of psychological distress during the COVID-19 pandemic [46]. In Japan, females are frequently engaged in the hospitality sector, including caregivers, nurses, reception workers, and cashiers; these kinds of gender roles may impact mental health.

(Page12, lines 308-312)

  1. Check word on line 303

Response and Modification: Thank you for pointing out the mistake: line 303 - “wav” should be “wave”. We revised the text as follows:

- The stress check was performed in 2020, the early stage of the third wave, and in 2021 after the fifth wave in Japan.

(Line 333)

  1. The conclusions seem straightforward; they should answer the question: has examining psychological distress only allowed you to discover that it has increased with the pandemic? What other data did you see?

Response: We thank the reviewer for the valuable suggestion and have added the following text in the conclusion section.

- Our results showed that the psychological distress of university staff increased slightly in 2020 and significantly in 2021. The long duration of the COVID-19 pandemic may disturb mental health. Time and various factors such as sex, occupation, and age had synergistic effects on mental health. Females and nurses were associated with psychological distress before the pandemic. Those who originally felt distressed were vulnerable to the negative effects of the pandemic on mental health. While carefully monitoring the impact of a prolonged pandemic on mental health, each workplace needs to consider specific interventions for those with high stress during the pandemic.

(Page12, lines 345-350)

In closing, Figure 1 allows us to see the differences per year in each variable more clearly.

Response: We appreciate your compliment. We have tried to make it easy to visually understand the changes over the three years using Figures.

Reviewer 3 Report

We welcome this original article and the idea of a longitudinal study on psychological distress before and during the COVID-19 pandemic.

The article is well structured, coherent, with a logical flow easy to navigate and uses appropriated references to the subject.

After checking for similarity with other sources (through Ithenticate), no significant similarities with other sources were identified.

From a methodological point of view, we suggest to the authors some details regarding the data collection procedure and insertion of ethical issues in data collection.

The results are well organized and logical, but we propose a more detailed description of them in sections 3.1. and 3.2.

In the conclusions section, we recommend the authors to insert some mentions regarding the practical applicability of the study results.

We want to congratulate you on a job well done!

Author Response

We greatly appreciate the review of our paper entitled “Psychological distress among university staff before and during the COVID-19 pandemic”; Manuscript ID: ijerph-2179558. The comments of the reviewers were helpful in improving and polishing our paper. We have revised our paper extensively according to the comments and marked up any revisions in the manuscript using the “Track Changes” function for better readability. Point by point systematic replies to the reviewers are below.

Reviewers' comments:

Reviewer #3: We welcome this original article and the idea of a longitudinal study on psychological distress before and during the COVID-19 pandemic.

The article is well structured, coherent, with a logical flow easy to navigate and uses appropriated references to the subject.

After checking for similarity with other sources (through Ithenticate), no significant similarities with other sources were identified.

Response: We appreciate the reviewers’ positive remarks and constructive comments. We have made revisions throughout the manuscript based on these comments.

Major Comments

  1. From a methodological point of view, we suggest to the authors some details regarding the data collection procedure and insertion of ethical issues in data collection.

Response and Modification:

We thank the reviewer for raising this important point. According to the suggestion, we have added the following text in the Methods.

- Information is gathered through online questionnaires and distribution of questionnaires. All participants were given the opportunity to refuse to enroll in our study.

(Page 2, lines 81-83)

One person was omitted because of refusal.

(Page 2, lines 86)

  1. The results are well organized and logical, but we propose a more detailed description of them in sections 3.1. and 3.2.

Response and Modification: We thank the reviewer for the valuable comment. We employed the use of bold font in several places to make it clearer in section 3.1. We revised the results section as follows:

- The scores of Stress Reaction in 2021 have increased significantly compared with 2019 and 2020; the results are similar based on each factor (Table S1).

(Page 8, lines 177-179, in section 3.1.)

- Following adjustment for confounding factors (age, sex, and occupation), the increase in scores of Stress Reaction was significantly associated with females and nurses over three years, before and during the COVID-19 pandemic (females, p < 0.001 in 2019, p < 0.001 in 2020, p < 0.001 in 2021; nurses, p = 0.012 in 2019, p < 0.001 in 2020, p < 0.001 in 2021) (Table 2). Teachers, nurses, allied health professionals, and doctors were related to elevated scores of Job Stressor during the 19 pandemic (teachers, p < 0.001 in 2020, p = 0.022 in 2021; nurses, p < 0.001 in 2020, p < 0.001 in 2021; allied health professional, p < 0.001 in 2020, p < 0.001 in 2021; doctors, p < 0.001 in 2020, p = 0.001 in 2021) (Table 2). Higher age led to increase in scores of Social Support, despite the pandemic (higher age, p < 0.001 in 2019, p < 0.001 in 2020, p < 0.001 in 2021) (Table 2).

Table S4 show the risk factors associated with the three components of the BJSQ. After adjusting for confounding factors (age, sex, occupation, and years), increased scores of Stress Reaction were associated significantly with females, nurses, and allied health professionals both before and during the COVID-19 pandemic (p < 0.001, p < 0.001, p = 0.046, respectively) (Table S4). Teachers, nurses, allied health professionals, and doctors were associated with increased scores of Job Stressor; the relation was stronger for HCWs than teachers, based on the standard partial regression coefficient (teachers [the standard partial regression coefficient = 0.0698/ p-value < 0.001]; nurses [0.3282/ p < 0.001]; allied health professionals [0.1465/ p < 0.001]; doctors [0.0856/ p < 0.001]) (Table S4). Social Support's standard partial regression coefficient for teachers was less than other occupations (teachers [the standard partial regression coefficient = − 0.0434/ p-value = 0.008]; nurses [0.0115/ p = 0.480]; allied health professionals [0.0370/ p = 0.011]; doctors [−0.0231/ p = 0.115]) (Table S4). Older age was found to be associated with decreases in scores of Stress Reaction and Job Stressor as well as an increase in scores of Social Support (Stress Reaction [the standard partial regression coefficient = − 0.0020/ p-value = 0.002]; Job Stressor [−0.0014/ p = 0.030]; Social Support [0.0072/ p < 0.001]) (Table S4).

(Page 8, lines 195-221, in section 3.2.)

  1. In the conclusions section, we recommend the authors to insert some mentions regarding the practical applicability of the study results.

Response and Modification: We thank the reviewer for the thoughtful suggestion. We have added our suggestion related to the practical applicability of our results in the conclusion section as follows:

- Our results showed that the psychological distress of university staff increased slightly in 2020 and significantly in 2021. The long duration of the COVID-19 pandemic may disturb mental health. Time and various factors such as sex, occupation, and age had synergistic effects on mental health. Females and nurses experienced psychological distress before the pandemic. Those who originally felt distressed were vulnerable to the negative effects of the pandemic on mental health. While carefully monitoring the impact of a prolonged pandemic on mental health, each workplace needs to consider specific interventions for those with high stress during the pandemic.

(Page 12, lines 348-350)
